# Role of Low Amount of Iron Intake from Groundwater for Prevention of Anemia in Children: A Cross-Sectional Study in Rural Bangladesh

**DOI:** 10.3390/nu16172844

**Published:** 2024-08-25

**Authors:** Sabuktagin Rahman, Patricia Lee, Nezam Uddin Biswas, Moududur Rahman Khan, Faruk Ahmed

**Affiliations:** 1Department of Public Health, American International University-Bangladesh, 408/1 Kuratoli, Khilkhet, Dhaka 1229, Bangladesh; 2School of Medicine and Dentistry, Public Health, Gold Coast Campus, Griffith University, Parklands Dr., Gold Coast, QLD 4222, Australia; patricia.lee@griffith.edu.au (P.L.); f.ahmed@griffith.edu.au (F.A.); 3Bangladesh National Nutrition Council, Mohakhali, Dhaka 1212, Bangladesh; nezam_ku@yahoo.com; 4Institute of Nutrition and Food Science, University of Dhaka, Dhaka 1000, Bangladesh; khan.moudud@gmail.com

**Keywords:** iron, groundwater, low-iron MNP, anemia, Bangladesh

## Abstract

In Bangladesh, groundwater, the principal source of drinking water, contains predominantly high levels of iron. Drinking groundwater is associated with good iron status in populations. Against this backdrop, iron supplementation is often associated with side effects, which reduces its intake compliance. However, the level of iron in groundwater is not consistent, and low levels exist in many areas of the country. In the present study, we examined the role of groundwater with a low concentration of iron in the prevention of anemia in Bangladeshi children. In 2018, a cross-sectional study was conducted in Bangladesh among children aged 2–5 years (n = 122) who drank groundwater containing a low level of iron (0–<2 mg/L). The combined intake of iron was calculated from the key sources—diet, groundwater, and the simulated intake of MNPs. The intakes of iron were compared against the standard reference intake. The children’s hemoglobin levels were measured using a photometer. The combined intake of iron from diet, groundwater with low levels of iron, and the simulated consumption of low-iron MNP in children was 5.8 ± 2.0 and 6.9 ± 2.5 mg/day, comprising 193% and 169% of the Estimated Average Requirements in the 2–3-year-old and 4–5-year-old subgroups, respectively. The combined intake of bioavailable iron from dietary and low-iron groundwater was 0.42 ± 0.023 and 0.22 ± 0.019 mg/day in children exposed to groundwater concentrations of 0.8–<2.0 mg/L and 0.0–<0.8 mg/L, respectively (*p* < 0.001). The mean concentration of hemoglobin in the respective groups was 12.17 ± 0.94 g/dL and 11.91 ± 0.91 g/dL (*p* = 0.30). The combined intake of iron from diet and the low-iron groundwater was associated with maintenance of hemoglobin concentration at the non-anemic level in > 90% of the children. The findings highlight the protective influence of the low concentration of iron in the drinking groundwater against childhood anemia in Bangladesh.

## 1. Introduction

In Bangladesh, the magnitude of anemia among children under five years old is high, with a range that varies from 33% to 51%, according to two nationally representative surveys [1,2]. Iron deficiency was thought to be the primary cause of anemia in the Bangladesh population; thus, a national policy for its prevention recommended iron supplementation, i.e., micronutrient powder (MNP) containing iron, for children aged 6–23 months [3,4]. Consequently, the MNP supplementation (containing 12.5 mg of iron) programs have been operational to prevent anemia in infants and children from 6 to 59 months old [5]. Nevertheless, the coverage of the MNP program is suboptimal [5], and compliance is poor because of iron-related side effects such as diarrhea, nausea, and vomiting [6].

However, contrary to popular perception, the National Micronutrient Survey 2011–2012 reported that the prevalence of iron deficiency (ID) in children under five years old was 10.7%, and iron deficiency anemia (IDA) was 7.2% [7]. Recent studies attributed the low prevalence of ID and IDA in the Bangladesh population to drinking iron-containing groundwater from tube wells [2,8]. Of note, groundwater (extracted from hand-pumped tube wells) is the principal source of drinking water for the large majority (97%) of the rural population in Bangladesh [9]. The hydrochemistry of Bangladesh groundwater reveals that it contains varying concentrations of dissolved iron, with a predominantly high concentration in many parts of Bangladesh, but a largely low concentration in other parts (Figure 1) of the country [2,9]. Furthermore, a prominent feature of groundwater is that the iron concentrations vary considerably between tube wells located near each other. Hence, even in predominantly high-iron groundwater areas, some wells contain a low concentration of iron in the water. 

Acknowledging the presence of a high concentration of iron in groundwater, the national anemia consultation of Bangladesh [3] recommended examining the efficacy and side effects of MNP supplementation with a low dose of iron for children residing in areas with predominantly high groundwater iron. Consequently, a recent trial conducted among Bangladeshi children aged 2–5 years demonstrated that low-iron MNP (containing 5 mg iron) was equally efficacious in preventing anemia compared to the standard MNP containing 12.5 mg of iron [10]. In addition, low-iron MNP was associated with significantly fewer incidences of key side effects, such as diarrhea, loose stools, nausea, and fever, compared to the standard MNP [10]. The study recommended low-iron MNP for the prevention of anemia in children under five years old residing in high-iron groundwater areas.

Given the geologically variable iron content in groundwater in Bangladesh, it would be difficult to introduce low-iron MNP (containing 5 mg iron) for children drinking high-iron groundwater and standard MNP (containing 12.5 mg iron) for children drinking low-iron groundwater. Administratively, it would be much more feasible and cost-effective to have a single composition (e.g., low-iron MNP) for the whole population. Therefore, the present cross-sectional study examined the following aspects: First, the role of drinking groundwater with a low level of iron (0–<2 mg/L) in relation to protection from childhood anemia; and secondly, the potential of the low-iron MNP in preventing childhood anemia in a predominantly low-iron groundwater setting. The specific objectives were as follows:a.To assess the combined intake of iron from the key sources (diet, low-iron-containing drinking groundwater, and simulated consumption of low-iron MNP).b.To compare the combined intake with the dietary reference intakes. c.To measure the concentration of hemoglobin in the children.d.To assess the association between combined iron intake from diet and low-iron-containing drinking groundwater with hemoglobin concentration in children.

The association of the combined iron consumption from diet plus low-iron drinking groundwater with hemoglobin level was examined with regard to the role of the low-iron groundwater in relation to protection from childhood anemia. On top of these sources, the hypothetical consumption of low-iron MNP was further examined with regard to the prospects of the supplement to control childhood anemia in this setting. 

## 2. Materials and Methods

### 2.1. Study Participants and Selection Process

The study participants were children aged 2–5 years residing in Belkuchi—a rural subdistrict of north-west Bangladesh. The children were selected from a household if the groundwater iron concentration of the tube wells they drank from was low, i.e., 0–<2 mg/L. Low concentration (<2 mg/L) is defined as per the cut-off recommended by the joint technical committee of the Food and Agricultural Organization and World Health Organization FAO/WHO [11]. The sampling was conducted over three stages (Figure 2). 

In the first stage, an inquiry was made with local residents to obtain information about the following: (1) households with a drinking water source (tube well) containing “low” or “no” iron; and (2) if the designated household had a child who met the age criteria. In the second stage, the initial verification of the iron content of groundwater was performed in the designated households by using a novel sensorial tool—taste-rating of a groundwater sample for the level of iron [12]. A tool with the standardized operational guidelines was used for qualifying the “taste rating” of the groundwater samples for a semi-quantitative assessment of iron content into three groups—“no iron”, “some iron”, and “heavy iron”. The details of the method are provided in Rahman et al. [12]. In case the taste rating suggested a “no” or “some” level of groundwater iron, in the final stage, a colorimetric device was used to confirm the stipulated concentration of iron in the groundwater (0–< 2 mg/L). A child was selected for the study if both conditions (age and level of iron in drinking water) were met. Data were collected from 1 August to 16 August 2018. 

### 2.2. Sample Size

There is a paucity of data on the intake of iron from drinking groundwater in children. The available study that estimated the intake of iron from groundwater reported a standard deviation (SD) of 6.5 [10]. That study was conducted considering tube wells with a high concentration of groundwater iron and had a mean concentration of iron ~8 mg/L (max. 43.3 mg/L). So, the wide range of the values of iron intake yielded a relatively large SD. The present study considered children who drank from groundwater with a concentration of iron 0–<2 mg/L. So, assuming a reasonably lower SD for the mean intake of iron from water in this population, a SD of 3.25, which is equivalent to 50% of the SD of the referred study, was considered reasonable in the present study. Allowing for a margin of error of 0.7 mg, with a variance (SD^2^) of 10.2, and at a 95% confidence level, the required sample size was 80, as per online sample size estimation software [13]. However, we measured the concentration of iron in the groundwater of 122 tube wells. Therefore, 122 children were interviewed and intake of iron from groundwater was assessed in the same number of children. 

To estimate the mean hemoglobin, considering the variance (SD^2^) of 0.608 [10], and an error margin of 0.15 g/dL at a 95% confidence level, the required sample size was 104 [13]. The actual number of the children who were measured for hemoglobin was 105.

### 2.3. Data Collection

#### 2.3.1. Assessment of Dietary Iron Intake 

The mother of the recruited child was asked about her child’s intake of food over the 24 h preceding the interview. The intake was captured by a 24 h recall over the six time prompts—breakfast, mid-morning, lunch, afternoon, dinner, and bedtime. The amount of the reported food intake was assessed by using food albums and utensils—plates, bowls, spoons, and packets/brands of the food items (i.e., processed foods). An updated food composition table on Bangladeshi foods [14] was used to calculate the intake of iron from the food items. 

#### 2.3.2. Assessment of Iron Intake from Groundwater (Drinking Water)

The concentration of iron in the drinking water sample was measured by a colorimetric test kit device (Hanna 3831, Hanna Instruments, Woonsocket, RI, USA) by using the manufacturer’s provided manual. To calculate the intake of iron from drinking groundwater, the total amount of water intake was first assessed over the 24 h preceding the interview using six time prompts following the methods described by Merrill et al. [8]. Intake of iron was calculated by multiplying the concentration of iron in the groundwater sample and the volume of intake of water over the 24 h preceding the interview [8,10]. 

#### 2.3.3. Intake of Iron from MNP Supplements

A simulated intake of the low-iron MNP was calculated. Since the compliance of MNP supplement consumption is generally suboptimal in the programmatic context of Bangladesh, the study considered the simulated intake of MNP representing two different compliance levels. Compliance levels of 85% and 50% were considered satisfactory [15] and suboptimal consumption, respectively. The intake of iron from the low-iron MNP was calculated by multiplying the dose (5 mg/day) by 0.85 (at 85% intake compliance) and 0.50 (at 50% intake compliance).

#### 2.3.4. Assessing the Total Iron Intake from Different Sources

The children’s intake of iron from drinking groundwater (i.e., tube wells) and their dietary intake of iron were estimated. Further, the simulated intake of iron from the low-iron MNPs was assessed. The combined intake of iron from different sources was calculated and compared with the dietary reference intakes, i.e., the Estimated Average Requirements (EAR) and the Recommended Dietary Allowance (RDA) for the stipulated age group of the children. The amount of bioavailable iron was calculated and compared with the reference values.

#### 2.3.5. Reference Standard for Comparison

The EAR and the RDA were used as references to assess the extent of iron intake from all sources. An EAR is the average daily dietary intake level sufficient to meet the nutrient requirements of half (50%) of the healthy individuals in a group. An RDA is the average daily dietary intake level sufficient to meet the nutrient requirements of nearly all (97–98%) of the healthy individuals in a group [16]. For children aged 2–3 years and 4–5 years, the RDAs of iron were 7 mg/day and 10 mg/day [16], while the EARs were 3 mg/day and 4.1 mg/day, respectively [17]. The intake of bioavailable iron was compared to the median and the 95th percentile of the joint FAO and WHO reference [18]. The medians were 0.46 mg/day and 0.50 mg/day in the 2–3-year-old and 4–5-year-old groups, respectively. The 95th percentiles were 0.58 mg/day and 0.63 mg/day in the respective age groups. 

#### 2.3.6. Assessing the Total Intake of Bioavailable Iron

The children’s age varied between 2 and 5 years; therefore, they had been exposed to iron from drinking water for different lengths of time. Additionally, the groundwater iron concentration of the tube wells could have been any value in the 0–2 mg/L range. Hence, taking these factors into consideration, it was assumed that some children might have had a depleted status of body iron, particularly the children who drank from the wells with zero or close to zero concentration of iron. Conversely, some children, principally those who drank from the tube wells with water iron concentration in the higher side of the 0–2 mg/L range, might have had repleted body-iron reserves. Therefore, to calculate the bioavailable iron from groundwater and present both scenarios, we used 40% efficiency of absorption of iron from the water considering iron-depleted participants and 10% considering iron-repleted participants, as reported by Worwood et al. [19]. Average absorption efficiency (23%) [19] was used in determining the amount of the bioavailable iron from the groundwater during the subgroup assessment (i.e., subgroups 0–<0.8 mg/L and ≥8–2.0 mg/L). This is because the range of the water iron concentration at the subgroup level would not have had a similar scale to the overall (all samples) range (0–2 mg/L). Assuming 40% absorption in the iron-depleted children (consuming from wells at a near-zero concentration of iron) and 10% absorption in the iron-repleted children (consuming from wells at near 2 mg/L iron concentration) is not applicable. Hence, the average absorption potential of 23% was considered at the subgroup level [19].

Regarding the absorption of iron from the MNP that contains ferrous fumarate as the iron complex and ascorbic acid as the absorption enhancer, for the participants who were iron-repleted, an absorption rate of 4.65% was considered [20]. For the participants with iron-depleted status, the absorption rate of 4.48% was considered [20]. The combined intake of bioavailable iron from all sources (dietary + groundwater + MNPs) was calculated considering the simulated usage of the low-iron MNP at the satisfactory (85%) [15] and suboptimal (50%) level of compliance. No children were consuming any iron supplements (including MNP) at the time of the study and in the preceding 6 months, and therefore such supplements were not considered for calculating the combined iron intakes. Regarding the bioavailability of dietary iron, taking into consideration the predominantly cereal-based traditional diet, a 5% absorption was considered [21].

Since one of the aspects of the study was the assessment of the potential of the low-iron MNP on hemoglobin status in a low-iron groundwater setting, an appraisal of the intake of iron from the key sources in conjunction with the simulated intake of low-iron MNP was provided in the results section. However, a similar assessment of the standard MNP, the existing MNP formulation in the country, was presented as Appendix A.

#### 2.3.7. Measurement of Hemoglobin

Following the collection of data on socio-demographics, dietary assessment, and iron intake from groundwater, the hemoglobin concentration of the children was measured via venous blood samples with a photometer (Hemocue 301, Hemocue AB, Angleholm Sweden). The blood was drawn from the median cubital vein with the participants seated on their mothers’ laps. Following proper asepsis of the puncture site with an alcohol pad, 0.5 mL of blood was drawn using a 3 mL disposable syringe. The blood sample was gently placed on a cover slip and sucked into a microcuvette of the photometer device. The measurement was performed by observing the manufacturer’s supplied manual.

### 2.4. Statistical Analysis

General characteristics of the children were estimated as mean ± SD and median (interquartile range) for the quantitative variables and proportion (%) for the categorical variables. The histogram visualization appeared nearly consistent with normality; however, the Shapiro–Wilk U coefficient, which is a stricter metric of normal distribution, was statistically significant. Shapiro-Wilk U test considered the variables such as children’s age (z = 2.27, *p* = 0.01), intake of groundwater iron (z = 4.65, *p* = 0.000), and intake of dietary iron (z = 5.35, *p* = 0.000). Statistical significance of the test was consistent with non-normal distribution; hence, the aggregated intake of iron from all the key sources over the preceding 24 h was presented as mean ± SD and median with interquartile ranges (IQRs). The simulated intake of iron from the standard (12.5 mg iron) and low-iron (5 mg iron) MNPs was deducted as the function of 85% (satisfactory compliance) [15] and 50% consumption (suboptimal compliance). Children were subgrouped by the median concentration (0.8 mg/L) of iron in their drinking water. The Mann–Whitney test was performed for the comparison of intakes of iron between the groups with *p*-values < 0.05 considered significant. However, since the hemoglobin concentrations were normally distributed (non-significant Shapiro–Wilk U coefficient), a Student’s *t*-test was performed to compare the groups. The prevalence of anemia between the groups was compared by a chi-square test. Linear regression was performed on the log-transformed data to assess the association of hemoglobin concentration vs. (a) the groundwater iron concentration, (b) the intake of groundwater iron, and (c) dietary iron. Additionally, linear models were depicted by graphs illustrating the 95% confidence intervals around the fitted lines. Furthermore, linear regression was performed on the log-transformed data to assess the association of the intake of groundwater iron and dietary iron with children’s hemoglobin concentration after adjusting for the groundwater iron concentration. Geometric coefficients were presented along with SE on the back-transformed data. All analyses were conducted with the statistical software STATA 17 (STATA Inc., College Station, TX, USA).

### 2.5. Ethical Considerations

The study was nested in a community-based trial examining the efficacy of a novel micronutrient powder formulation in children residing in areas with a high level of iron in groundwater. The trial received approval from the Research Ethical Committee of the Faculty of Biological Science, Dhaka University, Bangladesh (Ref# 46/Biol. Scs./2017–2018) and Griffith University Human Research Ethics Committee, Australia (Ref# 2017/467). Written informed consent was obtained from the participant’s (2–5 years old) guardian before the interview and blood collection. Data were kept in a secure place with the lead investigators. The participants remained anonymous during the presentation of the aggregated results. 

## 3. Results

Table 1 depicts the general characteristics of the children in the study. The mean ± SD age of the children was 3.58 ± 0.08 years. The proportion of female and male children was 44.3% and 55.7%, respectively. The common occupations of the household head were unskilled labor (32.8%), skilled labor (14.8%), business (12.3%), and farmer (11.5%). The mean ± SD duration of institutional education of the mothers was 6.3 ± 4.6 years. The mean ± SD weekly expense for purchasing basic food items was BDT 1801.3 ± 676.1 [USD 21.3 ± 8.0]. The mean ± SD amount of cultivable land possessed by households was 23.6 ± 43.5 decimals. The mean ± SD intake of water by the children over the preceding 24 h was 661.27 ± 73.1 mL and 955.40 ± 43.2 mL in 2–3-year-old and 4–5-year-old subgroups, respectively. 

Table 2 shows the mean ± SD intake of dietary iron, groundwater iron, and the combined intake from these sources by age groups of the children. The intake of dietary iron, groundwater iron, and the combined dietary and groundwater iron in children aged 2–3 years was 2.62 ± 1.84 mg/day, 0.64 ± 0.51 mg/day, and 3.3 ± 2.0 mg/day, respectively. The respective intakes in children aged 4–5 years were 3.51 ± 2.34 mg/day, 0.85 ± 0.73 mg/day, and 4.4 ± 2.5 mg/day. The combined intakes of dietary iron and water iron comprised 47.1% and 44% of the RDA in the respective age groups. The aggregated intakes exceeded the EAR for both the subgroups at 110% and 107.3%, respectively.

Table 3 depicts the intakes of actual and bioavailable iron from dietary and groundwater sources, the mean concentration of hemoglobin, and the prevalence of anemia in children, stratified by the median concentration of iron in groundwater (0.8 mg/L). The children’s mean ± SE intake of iron from drinking groundwater was 1.11 ± 0.07 mg/day and 0.24 ± 0.04 mg/day in the subgroups, defined as groundwater iron concentration (≥0.8–<2.0) mg/L and (0.0–<0.8) mg/L, respectively; (*p* < 0.001). The mean ± SE intake of the dietary iron between the subgroups did not differ statistically (3.32 ± 0.38 mg/day vs. 3.26 ± 0.23 mg/day, *p* = 0.79). The combined mean ± SE intake of bioavailable iron from dietary and groundwater sources was 0.42 ± 0.023 mg/day and 0.22 ± 0.019 mg/day in the subgroups (≥0.8–<2.0 mg/L) and (0.0–<0.8 mg/L), respectively; *p* < 0.001. The mean ± SE concentration of hemoglobin in children did not differ statistically (11.91 ± 0.91 g/dL (0.0–<0.8 mg/L subgroup) vs. 12.17 ± 0.94 g/dL (≥0.8–<2.0 mg/L subgroup), *p* = 0.30). The prevalence of anemia was 6.25% and 12.2% in the (≥0.8–<2.0 mg/L) and (0.0–<0.8 mg/L) subgroups, respectively (*p* = 0.29). 

Figure 3 shows the linear regression of the log-hemoglobin concentrations in children vs. (2A) log-groundwater iron concentration, (2B) log-intake of groundwater iron, and (2C) log-dietary intake of iron. The geometric mean of iron concentration of groundwater and the concentration of the children’s hemoglobin was positively associated at a 6% level of statistical significance; (coefficient: 1.08; *p* = 0.06, 2A). The geometric mean of the intake of groundwater iron and dietary iron vs. the hemoglobin concentrations were statistically significant (coefficient: 1.06, *p* = 0.03, 2B) and just significant (coefficient: 1.048, *p* = 0.05, 2C), respectively.

The linear regression (Table 4) showed that the log-intake of dietary iron is positively associated with log-hemoglobin concentration after adjusting for the log-groundwater iron concentration (Beta = 0.25; *p* = 0.023). After adjusting for the log-groundwater iron concentration, the log-intake of the water iron was associated with log-hemoglobin concentration at a 6% level of statistical significance (Beta 0.19, *p* = 0.06).

Table 5 presents the total intake of iron from all sources—dietary, groundwater, and the simulated intake of low-iron MNP—by children using the different levels of compliance, namely satisfactory (85%) and suboptimal (50%). The all source mean ± SD intakes of iron in children aged 2–3 years were 7.5 ± 2.0 mg/day and 5.8 ± 2.0 mg/day at the 85% and 50% compliance levels of MNP intakes, respectively. In the 4–5-year-old group, the respective intakes were 8.6 ± 2.5 mg/day and 6.9 ± 2.5 mg/day. At the satisfactory compliance of MNP (85%), all source intakes were 250% and 210% of the average reference intake (EAR) in the 2–3-year-old and 4–5-year-old groups, respectively. At the suboptimal compliance of MNP (50%), all source aggregated intakes were 193% and 169% of the EARs, respectively.

In the case of the intake of low-iron MNP at suboptimal compliance, the intake of iron from all sources was 82.8% and 69% of the RDAs in the respective age groups. In the case of satisfactory compliance, the RDAs exceeded at 107% and 86%, respectively (Table 5).

Table 6 presents the mean ± SD intake of bioavailable iron from all sources (dietary + groundwater + simulated intake of low-iron MNP) by the body iron status of the children, differential absorption potential of groundwater iron, and differential MNP compliance. In the case of 85% intake of low-iron MNP, mean ± SD and median (IQR) intake of bioavailable iron from the combined sources (diet, groundwater, low-iron MNP) are either on a par, higher or lower than the reference intakes of bioavailable iron [17] depending on the status of body iron. In 2–3-year-old iron-depleted children, the mean ± SD and median (IQR) intakes were 0.58 ± 0.24 mg/d and 0.49 (0.40–0.78) mg/d, respectively. In the case of the iron-repleted children, the mean ± SD and median (IQR) intakes of bioavailable iron were 0.39 ± 0.11 mg/d and 0.37 (0.31–0.45) mg/d, respectively, which are lower than the reference requirements-- median and the 95th percentile. At 50% intake of low-iron MNP, the intake of bioavailable iron in children generally was lower than the reference requirement irrespective of the iron status of the children, except the mean intake in the iron-depleted children, which was higher than the median of the reference.

In children aged 4–5 years, at 85% compliance of low-iron MNP, the intake of the combined sources of bioavailable iron was higher than the reference requirement except for the median intake relative to the 95th percentile of the requirement in children depleted of iron. In iron-repleted children, the intake was lower than reference requirements, i.e., the median and the 95th percentile.

### An Account of the Iron Intake in Relation to the Intake of the Standard MNP

In the case of the standard MNP, the estimated intake of iron from all sources exceeded the EAR considerably in both of the age groups; 365.8–463.3% (at the satisfactory compliance, 85%) and 258.5–316.6% (at the suboptimal compliance, 50%; Appendix A). The intake of bioavailable iron from all sources with the satisfactory MNP compliance exceeded both the median and the 95th percentile of the reference, irrespective of the iron status of the children. In the case of the suboptimal MNP compliance, the intake of bioavailable iron from all sources exceeded the median regardless of the iron status of the children and exceeded the 95th percentile in the iron-depleted children (Appendix A).

## 4. Discussion

The present study examined the intake of iron in 2–5-year-old rural Bangladeshi children from all the key sources, including the drinking of groundwater with a low concentration of iron (0–<2 mg/L), diet, and the simulated intake of low-iron MNP (or the standard MNP) at different compliance of consumption levels. Hemoglobin concentration of the children was measured to appraise a. the effect of combined intake of iron from the diet and the low-iron-containing drinking groundwater; and b. in addition to the diet and low-iron drinking water, the potential scope of low-iron MNP to prevent anemia in children in the setting.

Our results revealed that the combined intake of iron from diet and low-iron-containing drinking groundwater marginally (107–110%) exceeded the EAR in both age groups. However, when the simulated intake of low-iron MNP was added to that of the diet and groundwater, the intake relative to the EAR was further increased, e.g., 210–250% for satisfactory compliance and 168–193% for suboptimal compliance of MNP. The prevalence of anemia (in the absence of the simulated MNP intervention) was low (8.6%). 

Furthermore, the comparative intakes of groundwater with low levels of iron, dietary iron and hemoglobin concentrations in children sub-grouped by the median groundwater Fe concentration (0–<0.8 mg/L vs. ≥ 0.8–< 2 mg/L) were examined. Statistically, there was no difference in the concentrations of hemoglobin between the groups (*p* = 0.30). The statistical parity in hemoglobin concentrations between the groups was observed along with (a) the statistical parity in the amount of dietary iron and (b) about 4.5 times lower intake of groundwater iron in the group with groundwater iron concentration below the median (0–<0.8 mg/L). This demonstrated that even a very modest amount of daily consumption of iron from drinking water was associated with a fair concentration of hemoglobin at the non-anemic level. This is difficult to explain. Nonetheless, the observation compliments our finding that the intake of the groundwater iron and hemoglobin concentration of the children was positively associated (coefficient 1.06, *p* = 0.03). Further, when the linear associations of 1. the intake of groundwater iron and 2. the intake of dietary iron vs. the hemoglobin concentration were assessed after adjusting for the groundwater iron concentration, we found that the geometric coefficient for the intake of groundwater iron for a favorable influence on the hemoglobin concentration was 1.08 with a SE 1.04 (*p*-value 0.06, standardized Beta 0.19). The geometric coefficient for the intake of dietary iron for a favorable influence on the hemoglobin concentration was 1.06 with a SE 1.02 (*p*-value 0.023, standardized Beta 0.25).

The possible reason for the statistically significant *p*-value for the dietary iron predicting the hemoglobin concentration after adjusting for the groundwater iron concentration was perhaps the lower SE for the geometric coefficient. It suggests that the dietary intake of iron is relatively homogeneous across the children in the study and less variable than the intake of iron from groundwater. The intake of groundwater iron would be more variable over a wide range of iron concentrations (0–2 mg/L). Thus, the association of the intake of groundwater iron on hemoglobin concentration following the adjustment for groundwater iron concentration, despite a larger effect size (geometric coefficient), was constrained by a higher SE with a slight loss of precision, resulting in the *p*-value marginally outside 0.05. However, the judgment on the importance of a result needs to be based on the size of the effect seen and not the *p*-value only [22,23]. The thorough appraisal of the association of the groundwater iron intake and hemoglobin concentration through the linear regression with and without the adjustment for groundwater iron concentration provides some explanation that even a low intake of groundwater iron was compatible with the maintenance of hemoglobin concentration at the non-anemic level.

Further, the amount of the combined bioavailable iron from dietary and drinking water sources (0.22 mg/day) in children consuming very low amounts of iron from drinking water (iron concentration 0–<0.8 mg/L) was lower than the FAO/WHO recommended median of the daily requirement (0.46 mg/d) [17]. Despite the low amount of bioavailable iron, the mean concentration of hemoglobin in the group was 11.93 g/dL (median 12 g/dL), which is well above the cut-off for defining anemia (<11 g/dL), and most of the children (87.8%) were non-anemic. This is hard to explain on the back of low absorption (5%) of dietary iron due to their predominantly cereal-based food intake. We assume that despite being in small amounts, a constant daily dose of highly bioavailable (23% assumed in the present study) [19] iron from drinking water over the years might have replenished and developed the body’s iron reserve in the children to support hemoglobin synthesis. 

In relation to the bioavailability of iron, the aquifer of Bangladesh is inherently rich in soluble iron in ferrous form [24]. In Bangladesh, water is predominantly consumed within a short period of pumping, favoring the retention of bioavailable ferrous iron [24]. When water is stored in a container and consumed later, it is assumed that the ferrous iron might be oxidized to ferric form, rendering them to precipitate. Interestingly, the ferric iron is retained in solution as a colloid, being enabled by a high level of dissolved organic material (DOM) in the groundwater [25,26]. Moreover, after ingestion in the gastrointestinal tracts, duodenal cytochrome B (Dcytb), on the brush border of the enterocytes transforms the insoluble ferric iron into the absorbable ferrous form [27]. All of this evidence supports the high level of bioavailability of groundwater iron.

We observed that the association of groundwater iron concentration and the concentration of hemoglobin is further strengthened when cases with zero value of iron concentration in the drinking water are omitted. The zero values of the groundwater iron concentration were excluded to measure the association of the non-zero values with hemoglobin concentration. We assume that the non-zero concentration of iron in the drinking groundwater is likely to have a greater effect on hemoglobin than the zero-iron concentrations taking into consideration that the dietary intake of iron remains roughly similar. The linear regression of non-zero values of groundwater iron concentration vs. hemoglobin concentration was analyzed in the lower group (iron concentration (0–<0.8 mg group), which revealed that the regression coefficient was 12 (*p* = 0.037). With a small sample size with n = 21, the effect size is large and establishes the association.

We observed that the linear regression of the combined intake of iron from dietary and groundwater sources was positively associated with hemoglobin concentration with a statistical significance. When the simulated intake of low-iron MNP was added to the dietary and the water iron at either level of compliance, the association maintained the same magnitude of the coefficient with the same level of statistical significance. The reason is that the inclusion of the intake of the low-iron MNP was simulated; as such, a fixed mean amount of iron from MNP was added, which did not influence the coefficient from a statistical point of view. 

However, this does not preclude the prospect of the low-iron MNP for preventing childhood anemia in the setting. The salient findings of the study were as follows: dietary iron and water iron combined marginally exceeded the EARs (107–110%). When the simulated intake of iron from the low-iron MNP was added, the EARs were exceeded considerably (168–193%). Despite exceeding the EARs from the combined dietary and groundwater iron intake, about 9% of the children were anemic. Taking into consideration the bioavailability of iron, the combined amount of bioavailable iron just exceeded the median of the requirement if the low-iron MNP is added (at a suboptimal compliance) to the dietary and groundwater iron. The usage of the standard MNP at either level of compliance (satisfactory or suboptimal) might result in an even higher amount of bioavailable iron with potentially a higher likelihood of side effects. Furthermore, the groundwater iron concentrations vary considerably, and a very low level or “zero iron” (e.g., 16.4% in the present samples) exists in the groundwater of many areas of the country where dietary and water iron combined may not satisfactorily address anemia. Complementing this, the median intake of iron in children aged 2–3 years-old decreased by 45% (from 5.6 mg/day to 3.1 mg/day, Table 2 and Table 5) when the low-iron MNP at the suboptimal compliance is omitted from the combined all sources iron (diet, groundwater and low-iron MNP). Similarly, the median intake of iron in children aged 4–5 years-old decreased by 41% (from 6.2 mg/day to 3.67 mg/day, Table 2 and Table 5) when the low-iron MNP at the suboptimal compliance is taken off from the all sources iron. Thus, the observation further highlights the need for the low-iron MNP for the children to maximize the protection from anemia in this setting, particularly to the anemic children exposed to a very low concentration of water iron (i.e., closer to the zero iron values and/or very low intake of groundwater iron). Further study is required for the confirmation.

The strength of this study is that we used venous blood, which has a higher accuracy for the measurement of hemoglobin [28,29,30]; hence, the hemoglobin measured in the study better reflects the actual status. The study has some limitations. First, we did not consider the real-life consumption of MNP to account for its actual contribution to the intake of iron; instead, we used the simulated intake levels of MNPs for the determination of the combined intakes of iron from multiple sources. Second, consumed iron from multiple channels might have inter-source interactions in the gastrointestinal tract, which might influence the absorption in the intestines. Studying such interactions was beyond the scope of the study. Third, an iron status biomarker, e.g., ferritin, was not included in the study. This could have assessed the iron reserve to better explain the satisfactory level of hemoglobin in children, despite being exposed to a low amount of iron from groundwater. As per the raw data of the British Geological Survey 2001, very-low-iron-containing (<0.8 mg/L) groundwater represents around 40% of the tube wells in Bangladesh [9]. The present study sample may not be representative of the wider population from which they were drawn; thus, the results should be interpreted with caution. 

## 5. Conclusions

In conclusion, the study suggests that in low-iron groundwater settings in Bangladesh, the combined intake of iron from dietary and groundwater sources was associated with the maintenance of hemoglobin concentration at the non-anemic level in the large majority of the 2–5-year-old children. The small proportion of anemic children exposed to a very low level of iron from drinking groundwater might plausibly benefit from a low-dose iron supplement. A randomized controlled trial is required to confirm the findings.

## Figures and Tables

**Figure 1 nutrients-16-02844-f001:**
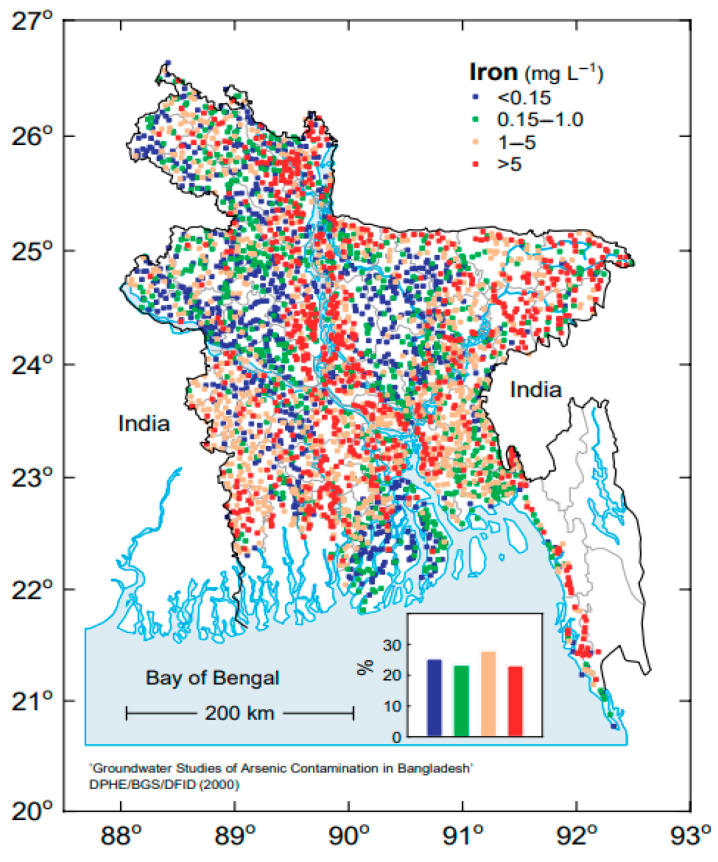
Variation in iron concentration in groundwater in different parts of Bangladesh [9] with permission of the BGS and DPHE, 2001. Arsenic contamination of groundwater in Bangladesh. Kinniburgh, D G and Smedley, P L (Editors). British Geological Survey Technical Report WC/00/19. British Geological Survey: Keyworth.

**Figure 2 nutrients-16-02844-f002:**
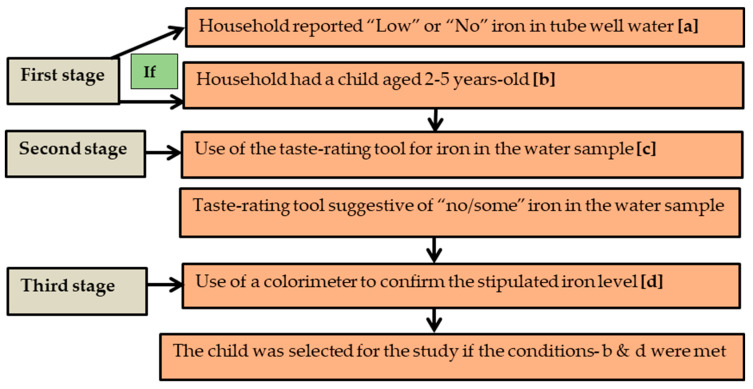
A schematic diagram showing the process of child selection.

**Figure 3 nutrients-16-02844-f003:**
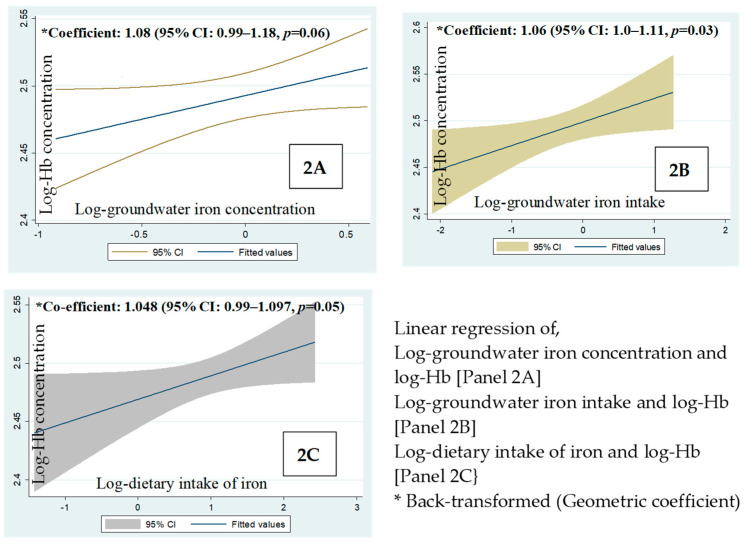
Linear regression of the log-hemoglobin concentration vs. (**2A**). Log-groundwater iron concentration, (**2B**). Log-intake of groundwater iron and (**2C**). Log-dietary iron.

**Table 1 nutrients-16-02844-t001:** Selected child and household characteristics.

Child and Household Characteristics	Estimates (N = 122)
Age (years)	3.58 ± 0.08
Sex (Female), n (%)	54 (44.3)
Sex (Male), n (%)	68 (55.74)
Occupation of household head	
Business, n (%)	15 (12.3)
Skilled labor, n (%)	18 (14.8)
Unskilled labor, n (%)	40 (32.8)
Farmer, n (%)	14 (11.5)
Mother’s education (years)	6.3 ± 4.6
Weekly expenses on principal food items * (BDT.) **	1801.3 ± 676.1
Amount of cultivable land (decimal unit)	23.6 ± 43.5
Intake of water (mL)	885.5 ± 428.4
Age 2–3 years (mL)	661.27 ± 73.1
Age 4–5 years (mL)	955.40 ± 43.2

Variables are reported as mean ± SD unless stated otherwise. * Rice, flour, oil, fish, meat, sugar, salt. ** Bangladeshi Taka.

**Table 2 nutrients-16-02844-t002:** Intake of dietary iron, groundwater iron, and combined intake in children by the age group.

Age SubGroup	RDA (mg/d)	EAR (mg/d)	Intake of Iron (Dietary) * (mg/d)		Intake of Iron (Water) † (mg/d)		Intake of Iron (dietary + Water) ‡ (mg/d)		Combined Intake % of Reference Intakes	
			**Mean ± SD** **(n)**	**Median** **(IQR)** **(n)**	**Mean** **±SD** **(n)**	**Median** **(IQR)** **(n)**	**Mean** **±SD** **(n)**	**Median** **(IQR)** **(n)**	**RDA (%)**	**EAR (%)**
2–3-year-old	7	3	2.62 ± 1.84 (31)	2.14 (1.48–3.22) (31)	0.64 ± 0.51 (31)	0.55 (0.26–1) (31)	3.3 ± 2.0 (31)	3.10 (1.84–4.26) (31)	47.1	110
4–5-year-old	10	4.1	3.51 ± 2.34 (91)	2.87 (1.77–4.6) (91)	0.85 ± 0.73 (91)	0.71 (0.31–1.26) (91)	4.4 ± 2.5 (91)	3.67 (2.37–5.92) (91)	44.0	107.3

RDA, Recommended Dietary Allowance; EAR, Estimated Average Requirements; SD, standard deviation; IQR, interquartile range. * Dietary iron is measured by a 24 h recall. † Intake of water iron is measured by a 24 h recall. ‡ Combined intake was calculated by the summation of intake of dietary iron and groundwater iron.

**Table 3 nutrients-16-02844-t003:** Comparative intakes of actual and bioavailable iron from groundwater and diet, and hemoglobin and anemia status in children in the subgroups by the median groundwater concentration.

Subgroups ∥	Intake of Water Iron (mg/day) Mean ± SE, n	*p*-Values
Groundwater iron concentration < 0.8 mg/L	0.24 ± 0.04 *, 44	<0.001
Groundwater iron concentration ≥ 0.8 mg/L	1.11 ± 0.07, 78	
Combined	0.80 ± 0.06, 122	
	Intake of dietary iron (mg/day) Mean ± SE, n	
Groundwater iron concentration < 0.8 mg/L	3.32 ± 0.38 *, 44	0.79
Groundwater iron concentration ≥ 0.8 mg/L	3.26 ± 0.23, 78	
Combined	3.28 ± 0.20, 122	
	Intake of bioavailable water iron (mg/day) § Mean ± SE, n	
Groundwater iron concentration < 0.8 mg/L	0.056 ± 0.009 *, 44	<0.001
Groundwater iron concentration ≥ 0.8 mg/L	0.26 ± 0.017, 78	
Combined	0.18 ± 0.014, 122	
	Bioavailable dietary iron (mg/day) Mean ± SE, n	
Groundwater iron concentration < 0.8 mg/L	0.16 ± 0.02 *, 44	0.79
Groundwater iron concentration ≥ 0.8 mg/L	0.16 ± 0.01, 78	
Combined	0.16 ± 0.11, 122	
	Bioavailable iron—groundwater and dietary combined (mg/day) Mean ± SE, n	
Groundwater iron concentration < 0.8 mg/L	0.22 ± 0.019 *, 44	<0.001
Groundwater iron concentration ≥ 0.8 mg/L	0.42 ± 0.023, 78	
Combined	0.35 ± 0.018, 122	
	Hemoglobin concentration (g/dL) Mean ± SE, n	
Groundwater iron concentration < 0.8 mg/L	11.91 ± 0.91 ^†^, 41	0.30
Groundwater iron concentration ≥ 0.8 mg/L	12.17 ± 0.94, 64	
Combined	12.07 ± 0.93, 105	
	Anemia, n (%)	
Groundwater iron concentration <0.8 mg/L	41 (12.2) ^‡^	0.29
Groundwater iron concentration ≥ 0.8 mg/L	64 (6.25)	
Combined	105 (8.6)	

* Mann–Whitney U test; ^†^ Student’s *t*-test; ^‡^ chi-square test; § Groundwater iron absorption 23%. ∥ Subgroups based on the median concentration of groundwater iron.; SE: Standard Error.

**Table 4 nutrients-16-02844-t004:** Linear regression showing associations of the 1. intake of groundwater iron and 2. dietary iron vs. hemoglobin concentration of the children after controlling for the groundwater iron concentration.

Independent Variables	Dependent Variable	Covariate	* Coefficient	SE	*p*-Value	Beta
I. Log-intake of groundwater iron	Log-Hb concentration	Log-groundwater iron concentration	1.08	1.04	0.06	0.19
II. Log-intake of dietary iron	Log-Hb concentration		1.06	1.02	0.023	0.25

* Coefficient of geometric mean (back-transformed); SE: Standard Error.

**Table 5 nutrients-16-02844-t005:** Estimates of the combined intake of iron from the key sources—dietary, groundwater, and the simulated intake of low-iron MNP—and the contribution to the dietary reference intakes at different compliance of the low-iron MNP.

Age Subgroup (Years), n	RDA mg/d	EAR mg/d	Intake of Iron (Dietary + Groundwater + MNP)				% of RDA * at 85% Compliance	% of EAR * at 85% Compliance	% of RDA * at 50% Compliance	% of EAR * at 50% Compliance
			Low-Iron MNP @ 85%		Low-Iron MNP @ 50%					
			Mean ±SD	Median (IQR)	Mean ±SD	Median (IQR)				
2–3, 31	7	3	7.5 ± 2.0	7.35 (6.1–8.5)	5.8 ± 2.0	5.6 (4.3–6.7)	107	250	82.8	193
4–5, 91	10	4.1	8.6 ± 2.5	7.92 (6.6–10.1)	6.9 ± 2.5	6.2 (4.9–8.4)	86	210	69	169

RDA, Recommended Dietary Allowance; EAR, Estimated Average Requirements; SD, standard deviation; IQR, interquartile range; MNP, micronutrient powder; * Proportions of RDA and EAR were calculated as the mean intake as a proportion of the RDA/EAR values.

**Table 6 nutrients-16-02844-t006:** Combined intake of bioavailable iron from the key sources along with the simulated intake of low-iron MNP at different compliance in iron-depleted and iron-repleted children.

Age Group (Year)	Absolute Requirement of Bioavailable Iron (mg/Day)		Bioavailable Iron from the Key Sources (Dietary + Groundwater + MNP)			
			Iron-Depleted Children Absorption of Groundwater Iron (40%) *		Iron-Repleted Children Absorption of Groundwater Iron (10%) *	
			**At 85% Intake of Low-Iron MNP**			
	**Median**	**95th Percentile**	**Mean ± SD, n** **mg/Day**	**Median (IQR), n** **mg/Day**	**Mean ± SD, n** **mg/Day**	**Median (IQR), n** **mg/Day**
2–3	0.46	0.58	0.58 ± 0.24, 31	0.49 (0.40–0.78), 31	0.39 ± 0.11, 31	0.37 (0.31–0.45), 31
4–5	0.50	0.63	0.71 ± 0.32, 91	0.61 (0.47–0.91), 91	0.46 ± 0.14, 91	0.42 (0.35–0.52), 91
2–5			0.67 ± 0.31, 122	0.58 (0.44–0.86), 122	0.44 ± 0.14, 122	0.40 (0.33–0.50), 122
			**At 50% intake of low-iron MNP**			
	**Median**	**95th Percentile**	**Mean ± SD, n** **mg/day**	**Median (IQR), n** **mg/day**	**Mean (SD), n** **mg/day**	**Median (IQR), n** **mg/day**
2–3	0.46	0.58	0.50 ± 0.24, 31	0.41 (0.32–0.70), 31	0.31 ± 0.11, 31	0.29 (0.23–0.37), 31
4–5	0.50	0.63	0.63 ± 0.32, 91	0.53 (0.39–0.84), 91	0.38 ± 0.14, 91	0.34 (0.27–0.44), 91
2–5			0.60 ± 0.31, 122	0.50 (0.36–0.79), 122	0.36 ± 0.14, 122	0.32 (0.25–0.42), 122

SD, standard deviation; IQR, interquartile range; MNP, micronutrient powder. * In the case of iron-depleted children, the absorption of iron from groundwater is 40%, and it is 10% in the case of iron-repleted children [19].

## Data Availability

The data sets used and/or analyzed during the current study are available from the corresponding author on reasonable request.

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
