# Peer review of "Role of Low Amount of Iron Intake from Groundwater for Prevention of Anemia in Children: A Cross-Sectional Study in Rural Bangladesh"

_nutrients, 2024, doi:10.3390/nu16172844_

Round 1
Reviewer 1 Report
Comments and Suggestions for Authors
Attached

Author Response
Point-by-point responses to the reviewer’s comments
Reviewer 1
Comments: The authors have obtained results about low-iron micronutrient powder (MNP) for prevention of anemia in children aged 2-5 years who drank groundwater containing a low level of iron but the authors only measured haemoglobin and should have measured other parameters related to iron metabolism such as ferritin concentration to know whether or not the iron stores in these children were depleted and there was a potential risk of iron deficiency anaemia. Really, the results of the study are inconsistent.
Response: Thank you for the important comment.
We acknowledge the reviewer’s observation. Ideally an iron status biomarker, such as ferritin could have been added to the understanding of the iron status, but we could not include it for budgetary constraint. Nonetheless, the evidence from the national micronutrient survey (Bangladesh National Micronutrient Survey 2011-12, Rahman et al, 2016) suggests that the Bangladeshi under-five children exposed to low level of iron in the drinking groundwater have a mean measure of inflammation-adjusted serum ferritin 23 ng/ml, which is much higher than the cut off defining ID in this age group (<12 ng/ml). This evidence serves as a proxy for our study, and we assume that the mean ferritin status in the children would likely be higher than the cut off of the deficiency.
It is to be noted that, in the study the actual intake of MNP was not measured. In the case of the actual consumption of the low iron MNP, it might have an incremental positive influence on ferritin on its own.
Iron from groundwater is highly bioavailable. The average absorption of groundwater iron is high at 23%. Since early months of life, a constant exposure to this highly bioavailable iron from the drinking water, steadily builds up iron reserve over the years. Iron is a principal element for synthesis of heme. The finding of our study is that the combined intake of dietary and the low level of groundwater iron were sufficient to keep >90% children protected from anemia, is a testament of the water-induced iron reserve in the body. All these observations suggest that iron-depletion in the children was unlikely in the setting.
Nonetheless, non-inclusion of ferritin is a weakness of the study, and we have duly acknowledged it in the manuscript (in page 17 in relation to limitations (highlighted).
Abstract
Comment: This sentence should be moved to the introduction or discussion section but not in the abstract.
“Because it has fewer side effects, a recent trial recommended low-iron micronutrient powder (MNP) for the prevention of anemia in Bangladeshi children exposed to a high concentration of iron from drinking groundwater.”
Response: Thanks for the suggestion. The sentence is removed from abstract.
Comment: This sentence should be moved to material and methods section “The children’s hemoglobin was measured using a photometer”
Response: Thanks for the comment. This is included in abstract as the key part of the methods. Moreover, the abstract reports the results of hemoglobin measurement, so keeping the Hb measurement in abstract is necessary.
Comments: Authors should indicate the number of the sample, the time of the trial…
Response: Sample size and the time of the study are now provided in abstract (highlighted).
Materials and methods
Comment: The authors only measured haemoglobin and should have measured other parameters related to iron metabolism such as ferritin concentration to know whether or not the iron stores in these children were depleted and there was a potential risk of iron deficiency anaemia.
Response: We acknowledge the reviewer’s observation. Ideally an iron status biomarker, such as ferritin could have been added to the understanding of the iron status, but we could not include it for budgetary constraint. Nonetheless, the evidence from the national micronutrient survey (Bangladesh National Micronutrient Survey 2011-12, Rahman et al, 2016) suggests that the Bangladeshi under-five children exposed to low level of iron in the drinking groundwater have a mean measure of inflammation-adjusted serum ferritin 23 ng/ml, which is much higher than the cut off defining ID in this age group (<12 ng/ml). This evidence serves as a proxy for our study, and we assume that the mean ferritin status in the children would likely be higher than the cut off of the deficiency.
Iron from groundwater is highly bioavailable. The average absorption of groundwater iron is high at 23%. Since early months of life, a constant exposure to this highly bioavailable iron from the drinking water, steadily builds up iron reserve over the years. Iron is a principal element for synthesis of heme. The finding of our study is that the combined intake of dietary and the low level of groundwater iron were sufficient to keep >90% children protected from anemia, is a testament of the water-induced iron reserve in the body. All these observations suggest that iron-depletion in the children was unlikely in the setting.
Nonetheless, non-inclusion of ferritin is a weakness of the study, and we have duly acknowledged it in the manuscript (in page 17 in relation to limitations (highlighted).
2.3.7. Measurement of haemoglobin. The authors should indicate at what point in the study the blood collection takes place.
Response: Measurement of hemoglobin was done, following the collection of data on socio-demographics, dietary assessment and iron intake assessment from groundwater. This is now mentioned in the methods section in relation to 2.3.7 (highlighted).
Results
Comment: The age of the study participants is 2-5 years but in table 1 they indicate that the mean age in months is 43.0±10.6 and this is not consistent.
Response: Thanks for the comment. The reason for using 2-5 years is to be consistent with the standard references of the dietary reference intakes (DRIs) which use age in years. Now in Table 1, the age is reported in years to be consistent (highlighted).
Table 1:
Comment:
- Why the authors it only includes the number of females and does not include the number of boys participating.
Response: Thanks for the comment. Now the boy’s proportion is presented in Table 1 (highlighted).
- Why the authors do not provide data for the 2-3- and 4-5-year age subgroups.
Response: Thanks for the comment. The reason for not segregating the age into subgroups to describe the socio-demographics is— The population is homogenous, so much so that segregating into subgroups (2-3 yrs & 4-5 yrs) shows very similar pattern of most of the socio-demographics such as occupation, education and household expenses on food. However, the intake of water is now presented segregated by age subgroups, since water intake increases with age (Highlighted in Table 1).
Comment
The authors indicate that “The prevalence of anemia was 6.25% and 12.2% in the (≥0.8-<2.0 mg/L) and (0.0-<0.8 mg/L)”
However, table 3 shows the number of individuals with anaemia for each group 41 and 64 and these percentages do not correspond:
Groundwater iron concentration <0.8mg/L 41(12.2)
Groundwater iron concentration >=0.8mg/L 64 (6.25)
Response: Thanks for the comment. To clarify, this subgrouping is done on the basis of the median concentration of groundwater iron (0.8 mg/L). The sample number in the subgroup <0.8 mg/L is 41; and that in the subgroup >=0.8 mg/L is 64. The prevalence of anemia in these subgroups is 12.2% and 6.25% respectively. It is worth noting that 41+64=105 children were considered for Hb measurement. Although 122 children were assessed for intake of iron from groundwater and dietary iron, Hb was measured in 105 children as per the sample size requirement. For details, please see the sample size section. Hope this is clarified.
Discussion
The discussion is not supported by relevant references, only 5-6 references appear:
- a) “However, the judgement on the importance of a result needs to be based on the size of the effect seen and not the p-value only [22, 23]”.
- b) “The strength of this study is that we used venous blood which has a higher accuracy for measurement of hemoglobin [24, 25, 26]”
Response: Thanks for the comment. The discussion part of the study was largely centered on the explanation of the findings. As the topic of groundwater iron and its relationship with anemia is a new evolving area of research, the relevant literature is sparse for comparison of the findings, which limited the number of references. However, we have expanded the discussion to elaborate the bioavailability of water iron as per the advice of another reviewer. That part of the discussion is supported by several additional references. (Highlighted in page 16 and the references 24-27).

Reviewer 2 Report
Comments and Suggestions for Authors
Authors present an interesting study on children from Bangladesh who drank water with different level of iron groundwater concentration. The study is cross-sectional, and investigates hypothetical MNP intake. I have some concerns:
1) sample size needs to be clearly stated in the abstract.
2) the title and abstract are somewhat misleading, as they imply that MNP was utilized in the study, whereas it was only modeled how it would affect the population. This needs to be more clearly stated. I suggest to remove MNP and include cross-sectional study in the title.
3) iron from inorganic compounds typically has low bioavailability, due to chemical form of ferric iron. The iron content of groundwater does not necessary come in same proportion of ferric /ferrous iron which can be commented in the discussion. Please include published data if known.
Author Response
Point-by-point responses to the reviewer’s comments
Reviewer 2
Comments and Suggestions for Authors
Authors present an interesting study on children from Bangladesh who drank water with different level of iron groundwater concentration. The study is cross-sectional and investigates hypothetical MNP intake. I have some concerns:
- sample size needs to be clearly stated in the abstract.
Response: Thanks for the comment. Sample size is mentioned in the abstract (highlighted).
2) the title and abstract are somewhat misleading, as they imply that MNP was utilized in the study, whereas it was only modeled how it would affect the population. This needs to be more clearly stated. I suggest removing MNP and include cross-sectional study in the title.
Response: Thank you for the very important comment. We have modified the title as suggested(highlighted). To complement the changes in the title, some adjustments are made in the abstract (highlighted), introduction (highlighted in page 3) and in discussion (highlighted in page 15).
3) iron from inorganic compounds typically has low bioavailability, due to chemical form of ferric iron. The iron content of groundwater does not necessary come in same proportion of ferric /ferrous iron which can be commented in the discussion. Please include published data if known.
Response: Thank you for the comment. Pertinent discussion is now included (page 16, highlighted)
That part of discussion is read as following in the discussion section of the manuscript,
“In relation to bioavailability of iron, aquifer of Bangladesh is inherently rich in soluble iron in ferrous form [24]. In Bangladesh, water is predominantly drunk within a short period of pumping favoring the retention of bioavailable ferrous iron [24]. When water is stored in a container and consumed later, it is assumed that the ferrous iron might be oxidized to ferric form, rendering them to precipitate. Interestingly, the ferric iron is retained in solution as a colloid, being enabled by high level of dissolved organic material (DOM) in the groundwater [25, 26]. Moreover, after the ingestion in the gastro-intestinal tracts, duodenal cytochrome B (Dcytb), on the brush border of the enterocytes transforms the insoluble ferric iron to absorbable ferrous form [27]. All this evidence supports the high level of bioavailability of groundwater iron.”

Round 2
Reviewer 1 Report
Comments and Suggestions for Authors
After reviewing the authors' responses and improvements to the manuscript, I recommend its publication.